# Hierarchical Decision Making by Generating and Following Natural Language Instructions

**Hengyuan Hu**[*]
Facebook AI Research
hengyuan@fb.com

**Denis Yarats**[*]
New York University & Facebook AI Research
denisyarats@cs.nyu.edu

**Qucheng Gong**
Facebook AI Research
qucheng@fb.com

**Yuandong Tian**
Facebook AI Research
yuandong@fb.com

**Mike Lewis**
Facebook AI Research
mikelewis@fb.com

## Abstract

We explore using natural language instructions as an expressive and compositional representation of complex actions for hierarchical decision making. Rather than directly selecting micro-actions, our agent first generates a plan in natural language, which is then executed by a separate model. We introduce a challenging real-time strategy game environment in which the actions of a large number of units must be coordinated across long time scales. We gather a dataset of 76 thousand pairs of instructions and executions from human play, and train *instructor* and *executor* models. Experiments show that models generate intermediate plans in natural langauge significantly outperform models that directly imitate human actions. The compositional structure of language is conducive to learning generalizable action representations. We also release our code, models and data[23].

## 1 Introduction

Many complex problems can be naturally decomposed into steps of high level planning and low level control. However, plan representation is challenging—manually specifying macro-actions requires significant domain expertise, limiting generality and scalability [18, 22], but learning composite actions from only end-task supervision can result in the hierarchy collapsing to a single action [3].

We explore representing complex actions as natural language instructions. Language can express arbitrary goals, and has compositional structure that allows generalization across commands [14, 1]. Our agent has a two-level hierarchy, where a high-level *instructor* model communicates a sub-goal in natural language to a low-level *executor* model, which then interacts with the environment (Fig. 1). Both models are trained to imitate humans playing the roles. This approach decomposes decision making into planning and execution modules, with a natural language interface between them.

We gather example instructions and executions from two humans collaborating in a complex game. Both players have access to the same partial information about the game state. One player acts as the *instructor*, and periodically issues instructions to the other player (the *executor*), but has no direct control on the environment. The *executor* acts to complete the *instruction*. This setup forces the *instructor* to focus on high-level planning, while the *executor* concentrates on low-level control.

---

[*]Equal Contribution.
[2]A demo is available at www.minirts.net
[3]Our code is open-sourced at www.github.com/facebookresearch/minirts

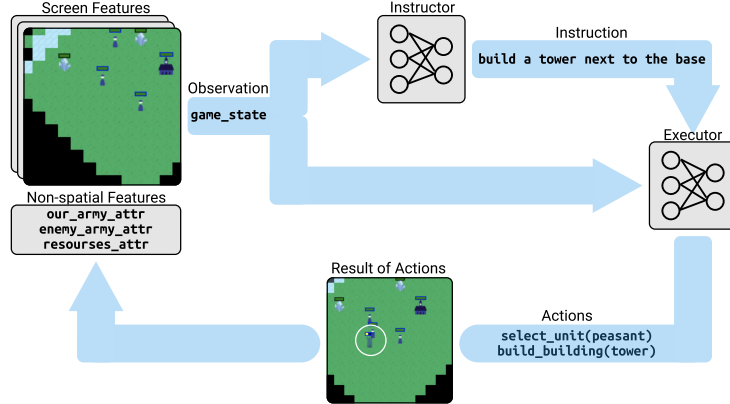

Figure 1: Two agents, designated *instructor* and *executor* collaboratively play a real-time strategy game (§2). The *instructor* iteratively formulates plans and issues instructions in natural language to the *executor*, who then executes them as a sequence of actions. We first gather a dataset of humans playing each role (§3). We then train models to imitate humans actions in each role (§4).

To test our approach, we introduce a real-time strategy (RTS) game, developing an environment based on [23]. A key property of our game is the *rock-paper-scissors* unit attack dynamic, which emphasises strategic planning over micro control. Our game environment is a challenging decision making task, because of exponentially large state-action spaces, partial observability, and the variety of effective strategies. However, it is relatively intuitive for humans, easing data collection.

Using this framework, we gather a dataset of 5392 games, where two humans (the *instructor* and *executor*) control an agent against rule-based opponents. The dataset contains 76 thousand pairs of human instructions and executions, spanning a wide range of strategies. This dataset poses challenges for both instruction generation and execution, as instructions may apply to different subsets of units, and multiple instructions may apply at a given time. We design models for both problems, and extensive experiments show that planning in language significantly improves performance.

In summary, we introduce a challenging RTS environment for sequential decision making, and a corresponding dataset of instruction-execution mappings. We develop novel model architectures with planning and control components, connected with a natural language interface. Agents that generate explicit natural language plans outperforms agents that directly imitate human actions, and we show that exploiting the compositional structure of language improves performance by allowing generalization across a large instruction set. We also release our code, models and data.

## 2   Task Environment

We implement our approach for an RTS game, which has several attractive properties compared to traditional reinforcement learning environments, such as Atari [13] or grid worlds [19]. The large state and action spaces mean that planning at different levels of abstraction is beneficial for both humans and machines. However, manually designed macro-actions typically do not match strong human performance, because of the unbounded space of possible strategies [21, 25]. Even with simple rules, adversarial games have the scope for complex emergent behaviour.

We introduce a new RTS game environment, which distills the key features of more complex games while being faster to simulate and more tractable to learn. Current RTS environments, such as StarCraft, have dozens of unit types, adding large overheads for new players to learn the game. Our new environment is based on MiniRTS [23]. It has a set of 7 unit types, designed with a *rock-paper-scissors* dynamic such that each has some units it is effective against and vulnerable to. Maps are randomly generated each game to force models to adapt to their environment as well as their opponent. The game is designed to be intuitive for new players (for example, catapults have long range and are effective against buildings). Numerous strategies are viable, and the game presents players with dilemmas such as whether to attack early or focus on resource gathering, or whether to commit to a

strategy or to attempt to scout for the opponent's strategy first. Overall, the game is easy for humans to learn, but challenging for machines due to the large action space, imperfect information, and need to adapt strategies to both the map and opponent. See the Appendix for more details.

## 3 Dataset

To learn to describe actions with natural language, we gather a dataset of two humans playing collaboratively against a rule-based opponent. Both players have access to the same information about the game state, but have different roles. One is designated the *instructor*, and is responsible for designing strategies and describing them in natural language, but has no direct control. The other player, the *executor*, must ground the instructions into low level control. The *executor*'s goal is to carry out commands, not to try to win the game. This setup causes humans to focus on either planning or control, and provides supervision for both generating and executing instructions.

We collect 5392 games of human teams against our bots.[4] Qualitatively, we observe a wide variety of different strategies. An average game contains 14 natural language instructions and lasts for 16 minutes. Each instruction corresponds to roughly 7 low-level actions, giving a challenging grounding problem (Table 1). The dataset contains over 76 thousand instructions, most of which are unique, and their executions. The diversity of instructions shows the wide range of useful strategies. The instructions contain a number of challenging linguistic phenomena, particularly in terms of reference to locations and units in the game, which often requires pragmatic inference. Instruction execution is typically highly dependent on context. Our dataset is available. For more details, refer to the Appendix.

| Statistic | Value |
|---|---|
| Total games | 5392 |
| Win rate | 58.6% |
| Total instructions | 76045 |
| Unique instructions | 50669 |
| Total words | 483650 |
| Unique words | 5007 |
| # words per instruction | 9.54 |
| # instructions per game | 14.1 |

Table 1: We gather a large language dataset for instruction generation and following. Major challenges include the wide range of unique instructions and the large number of low-level actions required to execute each instruction.

Analysing the list of instructions (see Appendix), we see that the head of the distribution is dominated by straightforward commands to perform the most frequent actions. However, samples from the complete instruction list reveal many complex compositional instructions, such as *Send one catapult to attack the northern guard tower [and] send a dragon for protection*. We see examples of challenging quantifiers (*Send all but 1 peasant to mine*), anaphora (*Make 2 more cavalry and send them over with the other ones*), spatial references (*Build a new town hall between the two west minerals patches*) and conditionals (*If attacked retreat south*).

## 4 Model

We factorize agent into an *executor* model (§4.2), which maps instructions and the game states into unit-level actions of the environment, and an *instructor* model (§4.3), which generates language instructions given the game states. We train both models with human supervision (§4.4).

### 4.1 Game Observation Encoder

We condition both the *instructor* and *executor* models on a fixed-sized representation of the current game state, which we construct from a spatial map observation, internal states of visible units, and several previous natural language instructions. (Fig. 2). We detail each individual encoder below.

### 4.1.1 Spatial Inputs Encoder

We encode the spatial information of the game map using a convolutional network. We discretize the map into a $32 \times 32$ grid and extract different bits of information from it using separate channels. For example, three of those channels provide binary indication of a particular cell visibility, which

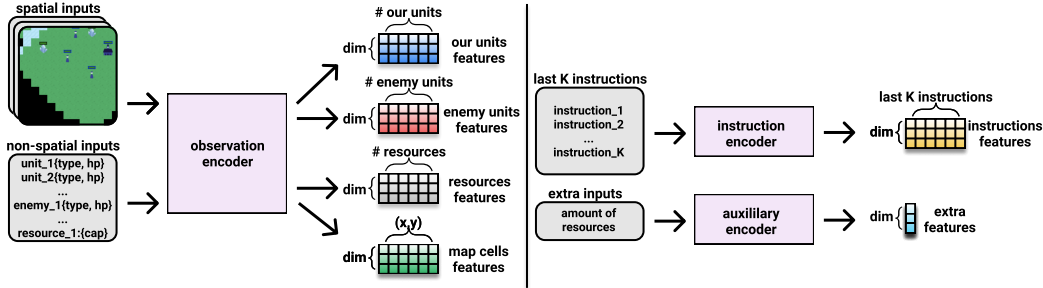

Figure 2: At each time step of the environment we encode spatial observations (e.g. the game map) and non-spatial internal states for each game object (e.g. units, buildings, or resources) via the observation encoder, which produces separate feature vectors for each unit, resource, or discrete map locations. We also embed each of the last $K$ natural language instructions into individual instruction feature vectors. Lastly, we learn features for all the other global game attributes by employing the auxiliary encoder. We then use these features for both the *executor* and *instructor* networks.

indicates INVISIBLE, SEEN, and VISIBLE states. We also have a separate channel per unit type to record the number of units in each spatial position for both our and enemy units separately. Note that due to "fog-of-war", not all enemy units are visible to the player. See the Appendix for more details.

We apply several $3 \times 3$ convolutional layers that preserve the spatial dimensions to the input tensor. Then we use 4 sets of different weights to project the shared convolutional features onto different 2D features spaces, namely OUR UNITS, ENEMY UNITS, RESOURCES, and MAP CELLS. We then use $(x, y)$ locations for units, resources, or map cells to extract their features vectors from corresponding 2D features spaces.

### 4.1.2 Non-spatial Inputs Encoder

We also take advantage of non-spatial attributes and internal state for game objects. Specifically, we improve features vectors for OUR UNITS and ENEMY UNITS by adding encodings of units health points, previous and current actions. If an enemy unit goes out the players visibility, we respect this by using the state of the unit's attributes from the last moment we saw it. We project attribute features onto the same dimensionality of the spatial features and do a element-wise multiplication to get the final set of OUR UNITS and ENEMY UNITS features vectors.

### 4.1.3 Instruction Encoders

The state also contains a fixed-size representation of the current instruction. We experiment with:

- An instruction-independent model (EXECUTORONLY), that directly mimics human actions.
- A non-compositional encoder (ONEHOT) which embeds each instruction with no parameter sharing across instructions (rare instructions are represented with an *unknown* embedding).
- A bag-of-words encoder (BOW), where an instruction encoding is a sum of word embeddings. This model tests if the compositionality of language improves generalization.
- An RNN encoder (RNN), which is order-aware. Unlike BOW, this approach can differentiate instructions such as *attack the dragon with the archer* and *attack the archer with the dragon*.

### 4.1.4 Auxiliary Encoder

Finally, we encode additional game context, such as the amount of money the player has, through a simple MLP to get the EXTRA features vector.

### 4.2 Executor Model

The *executor* predicts an action for every unit controlled by the agent based on the global summary of the current observation. We predict an action for each of the player's units by choosing over an

ACTION TYPE first, and then selecting the ACTION OUTPUT. There are 7 action types available: IDLE, CONTINUE, GATHER, ATTACK, TRAIN UNIT, BUILD BUILDING, MOVE. ACTION OUTPUT specifies the target output for the action, such as a target location for the MOVE action, or the unit type for TRAIN UNIT. Fig. 3 gives an overview of the *executor* design, also refer to the Appendix.

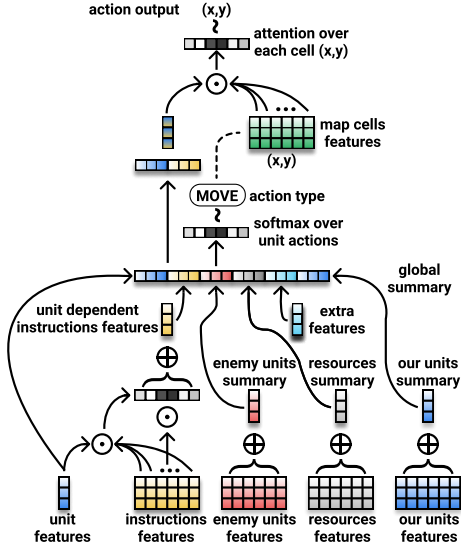

For each unit, we consider a history of recent $N$ instructions ($N = 5$ in all our experiments), because some units may still be focusing on a previous instruction that has long term effect like *keep scouting* or *build 3 peasants*. To encode the $N$ instructions, we first embed them in isolation with the 4.1.3. We take $K$ that represents how many frames have passed since that instruction gets issued and compute $H = \max(H_{max}, K/B)$ where $H_{max}, B$ are constants defining the number of bins and bin size. We also take $O = 1, 2, ..., N$ that represents the temporal ordering of those instructions. We embed $O$ and $H$ and concatenate the embeddings with language embedding. Dot product attention is used to compute an attention score between a unit and recent instructions and then a unit dependent instruction representation is obtained through a weighted sum of history instruction embeddings using attention score as weight.

We use the same observation encoder (§4.1) to obtain the features mentioned above. To form a global summary, we sum our unit features, enemy unit features, and resource features respectively and then concatenate together with EXTRA features.

Figure 3: Modeling an action for an unit requires predicting an action type based on the **global summary** of current observation, and then, depending on the predicted action type, computing a probability distribution over a set of the action targets. In this case, the MOVE action is sampled, which uses the map cells features as the action targets.

To decide the action for each unit, we first feed the concatenation of the unit feature, unit depending instruction feature and the global summary into a multi-layer neural classifier to sample an ACTION TYPE. Depending on the action type, we then feed inputs into different action-specific classifiers to sample ACTION OUTPUT. In the action argument classifier, the unit is represented by the concatenation of unit feature and instruction feature, and the targets are represented by different target embeddings. For ATTACK, the target embeddings are enemy features; for GATHER; the target embeddings are resource features; for MOVE, the target embeddings are map features; for TRAIN UNIT, the target embeddings are embeddings of unit types; for BUILD BUILDING, the target embeddings are embeddings of unit types and map features, and we sample type and location independently. The distribution over targets for the action is computed by taking the dot product between the unit representation and each target, followed by a softmax.

We add an additional binary classifier, GLOBAL CONTINUE, that takes the global summary and **current** instruction embedding as an input to predict whether all the agent's units should continue working on their previous action.

## 4.3 Instructor Model

The *instructor* maps the game state to instructions. It uses the game observation encoder (§4.1) to compute a global summary and **current** instruction embedding similar to the *executor*. We experiment with two model types:

**Discriminative Models** These models operate on a fixed set of instructions. Each instruction is encoded as a fixed-size vector, and the dot product of this encoding and the game state encoding is fed into a softmax classifier over the set of instructions. As in §4.1.3, we consider non-compositional (ONEHOT), bag-of-words (BOW) and RNN DISCRIMINATIVE encoders.

**Generative Model** The discriminative models can only choose between a fixed set of instructions. We also train a generative model, RNN GENERATIVE, which generates instructions autoregressively. To compare likelihoods with the discriminative models, which consider a fixed set of instructions, we re-normalize the probability of an instruction over the space of instructions in the set.

The *instructor* model must also decide at each time-step whether to issue a new command, or leave the *executor* to follow previous instructions. We add a simple binary classifier that conditions on the global feature, and only sample a new instruction if the result is positive.

Because the game is only partially observable, it is important to consider historical information when deciding an instruction. For simplicity, we add a running average of the number of enemy units of each type that have appeared in the visible region as an extra input to the model. To make the *instructor* model aware of how long the current instruction has been executed, we add an extra input representing number of time-step passed since the issuance of current instruction. As mentioned above, these extra inputs are fed into separate MLPs and become part of the EXTRA feature.

## 4.4  Training

Since one game may last for tens of thousands of frames, it is not feasible nor necessary to use all frames for training. Instead, we take one frame every $K$ frames to form the supervised learning dataset. To preserve unit level actions for the *executor* training, we put all actions that happen in $[tK, (t+1)K)$ frames onto the $tK$th frame if possible. For actions that cannot happen on the $tK$th frame, such as actions for new units built after $tK$th frame, we simply discard them.

Humans players sometimes did not execute instructions immediately. To ensure our *executor* acts promptly, we filter out action-less frames between a new instruction and first new actions.

### 4.4.1  Executor Model

The *executor* is trained to minimize the following negative log-likelihood loss:

$$\mathcal{L} = -\log P_{\text{cont}}(c|s) - (1-c) \cdot \sum_{i=1}^{|u|} \log P_{\text{A}}(a_{u_i}|s)$$

where $s$ represents game state and instruction, $P_{\text{cont}}(\cdot|s)$ is the *executor* GLOBAL CONTINUE classifier (see §4.2), $c$ is a binary label that is 1 if all units should continue their previous action, $P_A(a_{u_i}|s)$ is the likelihood of unit $i$ doing the correct action $a_{u_i}$.

### 4.4.2  Instructor Model

The loss for the *instructor* model is the sum of a loss for deciding whether to issue a new instruction, and the loss for issuing the correct instruction:

$$\mathcal{L} = -\log P_{\text{cont}}(c|s) - (1-c) \cdot \mathcal{L}_{\text{lang}}$$

where $s$ represents game state and current instruction, $P_{\text{cont}}(\cdot|s)$ is the continue classifier, and $c$ is a binary label with $c = 1$ indicating that no new instruction is issued. The language loss $\mathcal{L}_{\text{lang}}$ is the loss for choosing the correct instruction, and is defined separately for each model.

For ONEHOT instructor, $\mathcal{L}_{\text{lang}}$ is simply negative log-likelihood of a categorical classifier over a pool of $N$ instructions. If the true target is not in the candidate pool $\mathcal{L}_{\text{lang}}$ is 0.

Because BOW and RNN DISCRIMINATIVE can compositionally encode any instruction (in contrast to ONEHOT), we can additionally train on instructions from outside the candidate pool. To do this, we encode the true instruction, and discriminate against the $N$ instructions in the candidate pool and another $M$ randomly sampled instructions. The true target is forced to appear in the $M + N$ candidates. We then use the NLL of the true target as language loss. This approach approximates the expensive softmax over all 40K unique instructions.

For RNN GENERATIVE, the language loss is the standard autoregressive loss.

| Executor Model | Negative Log Likelihood | Win/Lose/Draw Rate (%) |
|---|---|---|
| EXECUTORONLY | $3.15 \pm 0.0024$ | 41.2/40.7/18.1 |
| ONEHOT | $3.05 \pm 0.0015$ | 49.6/37.9/12.5 |
| BOW | $2.89 \pm 0.0028$ | 54.2/33.9/11.9 |
| RNN | $\mathbf{2.88 \pm 0.0006}$ | **57.9/30.5/11.7** |

Table 2: Negative log-likelihoods of human actions for *executor* models, and win-rates against EXECUTORONLY (which does not use an instructor model to generate natural language plans). We use the RNN DISCRIMINATIVE *instructor* with 500 instructions. Modelling instructions compositionally improves performance, showing linguistic structure enables generalization.

| Instructor Model (with N instructions) | Negative Log Likelihood | | | Win/Lose/Draw rate (%) | | |
|---|---|---|---|---|---|---|
| | N=50 | N=250 | N=500 | N=50 | N=250 | N=500 |
| ONEHOT | $0.662 \pm 0.005$ | $0.831 \pm 0.001$ | $0.911 \pm 0.005$ | 44.6 / 43.4 / 12.0 | 49.7 / 35.9 / 14.3 | 43.1 / 41.1 / 15.7 |
| BOW | $0.638 \pm 0.004$ | $0.792 \pm 0.001$ | $0.869 \pm 0.002$ | 41.3 / 41.2 / 17.5 | 51.5 / 33.3 / 15.3 | 50.5 / 37.1 / 12.5 |
| RNN DISCRIMINATIVE | $\mathbf{0.618 \pm 0.005}$ | $\mathbf{0.764 \pm 0.002}$ | $\mathbf{0.826 \pm 0.002}$ | 47.8 / 36.5 / 15.7 | 55.4 / 33.1 / 11.5 | **57.9 / 30.5 / 11.7** |
| RNN GENERATIVE | $0.638 \pm 0.006$ | $0.794 \pm 0.006$ | $0.857 \pm 0.002$ | 47.3 / 38.1 / 14.6 | 51.1 / 33.7 / 15.2 | 54.8 / 33.8 / 11.4 |

Table 3: Win-rates and likelihoods for different *instructor* models, with the $N$ most frequent instructions. Win-rates are against a non-hierarchical *executor* model, and use the RNN *executor*. Better results are achieved with larger instruction sets and more compositional instruction encoders.

# 5 Experiments

We compare different *executor* (§5.1) and *instructor* (§5.2) models in terms of both likelihoods and end-task performance. We show that hierarchical models perform better, and that the compositional structure of language improves results by allowing parameter sharing across many instructions.

## 5.1 Executor Model

The *executor* model learns to ground pairs of states and instructions onto actions. With over 76 thousand examples, a large action space, and multiple sentences of context, this problem in isolation is one of the largest and most challenging tasks currently available for grounding language in action.

We evaluate *executor* performance with different instruction encoding models (§4.1.3). Results are shown in Table 2, and show that modelling instructions compositionally—by encoding words (BOW) and word order (RNN)—improves both the likelihoods of human actions, and win-rates over non-compositional *instructor* models (ONEHOT). The gain increases with larger instruction sets, demonstrating that a wide range of instructions are helpful, and that exploiting the compositional structure of language is crucial for generalization across large instruction sets.

We additionally ablate the importance of considering multiple recent instructions during execution (our model performs attention over the most recent 5 commands §4.2). When considering only the current instruction with the RNN *executor*, we find performance drops to a win-rate of 52.9 (from 57.9) and negative log likelihood worsens from 2.88 to 2.93.

## 5.2 Instructor Model

We compare different *instructor* models for mapping game states to instructions. As in §5.1, we experiment with non-compositional, bag-of-words and RNN models for instruction generation. For the RNNs, we train both a discriminative model (which maps complete instructions onto vectors, and then chooses between them) and a generative model that outputs words auto-regressively.

Evaluating language generation quality is challenging, as many instructions may be reasonable in a given situation, and they may have little word overlap. We therefore compare the likelihood of the human instructions. Our models choose from a fixed set of instructions, so we measure the likelihood of choosing the correct instruction, normalized over all instructions in the set. Likelihoods across different instructions sets are not comparable.

Table 3 shows that, as §5.1, more structured instruction models give better likelihoods—particularly for larger instruction sets, which are harder to model non-compositionally.

We compare the win-rate of our models against a baseline which directly imitates human actions (without explicit plans). All models that plan by generating and executing instructions outperform this baseline. More compositional instruction encoders improve performance, and can use more instructions effectively. These results demonstrate the potential of language for compositionally representing large spaces of complex plans.

## 5.3 Qualitative Analysis

Observing games played by our model, we find that most instructions are both generated and executed as humans plausibly would. The *executor* is often able to correctly count the number of units it should create in commands such as *build 3 dragons*.

There are several limitations. The *executor* sometimes acts without instructions—partly due to mimicking some humans behaviour, but also indicating a failure to learn dependencies between instructions and actions. The *instructor* sometimes issues commands which are impossible in its state (e.g. to attack with a unit that the it does not have)—causing weak behaviour from *executor* model.

## 6 Related work

Previous work has used language to specify exponentially many policies [14, 1, 27], allowing zero-shot generalization across tasks. We develop this work by generating instructions as well as executing them. We also show how complex tasks can be decomposed into a series of instructions and executions.

Executing natural language instructions has seen much attention. The task of grounding language into an executable representation is sometimes called semantic parsing [29], and has been applied to navigational instruction following, e.g. [2]. More recently, neural models instruction following have been developed for a variety of domains, for example [11] and [10]. Our dataset offers a challenging new problem for instruction following, as different instructions will apply to different subsets of available units, and multiple instructions may be apply at a given time.

Instruction generation has been studied as a separate task. [7] map navigational paths onto instructions. [8] generate instructions for complex tasks that humans can follow, and [9] train a model for instruction generation, which is used both for data augmentation and for pragmatic inference when following human-generated instructions. We build on this work by also generating instructions at test time, and showing that planning in language can improve end task performance.

Learning to play a complete real-time strategy game, including unit building, resources gathering, defence, invasion, scouting, and expansion, remains a challenging problem [15], in particular due to the complexity and variations of commercially successful games (e.g., StarCraft I/II), and its demand of computational resources. Traditional approaches focus on sub-tasks with hand-crafted features and value functions (e.g., building orders [5], spatial placement of building [4], attack tactics between two groups of units [6], etc). Inspired by the recent success of deep reinforcement learning, more works focus on training a neural network to finish sub-tasks [24, 17], some with strong computational requirement [28]. For full games, [23] shows that it is possible to train an end-to-end agent on a small-scaled RTS game with predefined macro actions, and TStarBot [20] applies this idea to StarCraft II and shows that the resulting agent can beat carefully-designed, and even cheating rule-based AI. By using human demonstrations, we hand crafting macro-actions.

Learning an end-to-end agent that plays RTS games with unit-level actions is even harder. Progress is reported for MOBA games, a sub-genre of RTS games with fewer units—for example, [16] shows that achieving professional level of playing DoTA2 is possible with massive computation, and [26] shows that with supervised pre-training on unit actions, and hierarchical macro strategies, a learned agent on Honor of Kings is on par with a top 1% human player.

## 7 Conclusion

We introduced a framework for decomposing complex tasks into steps of planning and execution, connected with a natural language interface. We experimented with this approach on a new strategy game which is simple to learn but features challenging strategic decision making. We collected a

large dataset of human instruction generations and executions, and trained models to imitate each role. Results show that leveraging the compositional structure of natural language benefits generalization for both the *instructor* and *executor* model, outperforming agents that do not plan with language. Future work should use reinforcement learning to further improve the planning and execution models, and explore generating novel instructions.

## Footnotes

[4]Using ParlAI [12]

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
