[Supplementary Material]

# Hierarchical Decision Making by Generating and Following Natural Language Instructions

## A Detailed game design

We develop an RTS game based on the MiniRTS framework, aspiring to make it intuitive for humans, while still providing a significant challenge to machines due to extremely high-dimensional observation and actions spaces, partial observability, and non-stationary environment dynamics imposed by the opponent. Below we describe the key game concepts.

### A.1 Game units specifications

**Building units** Our game supports 6 different building types, each implementing a particular function in the game. Any building unit can be constructed by the PEASANT unit type at any available map location by spending a specified amount of resources. Later, the constructed building can be used to construct units. Most of the building types can produce up to one different unit type, except of WORKSHOP, which can produce 3 different unit types. This property of the WORKSHOP building allows various strategies involving bluffing. A full list of available building units can be found in Table 1.

**Army units** The game provides a player with 7 army unit types, each having different strengths and weaknesses. PEASANT is the only unit type that can construct building units and mine resources, so it is essential for advancing to the later stages of the game. We design the attack relationships between each unit type with a *rock-paper-scissors* dynamic—meaning that each unit type has another unit type that it is effective against or vulnerable to. This property means that effective agents must be reactive to their opponent's strategy. See Fig. 1 for a visualization. Descriptions of army units can be found in Table 2.

Figure 1: Our game implements the *rock-paper-scissors* attack graph, where each unit has some units it is effective against and vulnerable to.

**Resource unit** RESOURCE is a stationary and neutral unit type, it cannot be constructed by anyone, and is only created during the map generation phase. PEASANTs of both teams are allowed to mine the same RESOURCE unit, until it is exhausted. Initial capacity is set to 500, and one mine action subtracts 10 points from the RESOURCE. Several RESOURCE units are placed randomly on the map, which gives raise to many strategies around RESOURCE domination.

## A.2   Game map

We represent the game map as a discrete grid of 32x32. Each cell of the grid can either be grass or water, where the grass cell is passable for any army units, while the water cell prevents all units except of DRAGON to go through. Having water cells around one's main base can be leveraged as a natural protection. We generate maps randomly for each new game, we first place one TOWN HALL for each player randomly. We then add some water cells onto the map, making sure that there is at least one path between two opposing TOWN HALLs, but otherwise aiming to create bottlenecks. Finally, we randomly locate several RESOURCE units onto the map such that they are approximately equidistant from the players TOWN HALLs.

## B   RTS game as an Reinforcement Learning environment

Our platform can be also used as an RL environment. In our code base we implement a framework that allows a straightforward interaction with the game environment in a canonical RL training loop. Below we detail the environment properties.

### B.1   Observation space

We leverage both spatial representation of the map, as well as internal state of the game engine (e.g. units health points and attacking cool downs, the amount of resources, etc.) to construct an observation. We carefully address the fog of war, by masking out the regions of the map that have not been visited. In addition, we remove any unseen enemy units attributes from the observation. The partial observability of the environment makes it especially challenging to apply RL due to highly non-stationary state distribution.

### B.2   Action space

At each timestep of the environment we predict an action for each of our units, both buildings and army. The action space is consequently large—for example, any unit can go to any location at each timestep. Prediction of an unit action proceeds in steps, we first predict an action type (e.g. MOVE or ATTACK), then, based on the action type, we predict the action outputs. For example, for the BUILD BUILDING action type the outputs will be the type of the future building and its location on the game map. We summarize all available action types and their structure in Table 3.

### B.3   Reward structure

We support a sparse reward structure, e.g. the reward of 1 is issued to an agent at the end if the game is won, all the other timesteps receive the reward of 0. Such reward structure makes exploration an especially challenging given the large dimensionality of the action space and the planning horizon.

## C   Data collection

We design a data collection task based on ParlAI, a transparent framework to interact with human workers. We develop separate game control interfaces for both the *instructor* and the *executor* players,

| Building name | Description |
|---|---|
| TOWN HALL | The main building of the game, it allows a player to train PEASANTs and serves as a storage for mined resources. |
| BARRACK | Produces SPEARMEN. |
| BLACKSMITH | Produces SWORDMEN. |
| STABLE | Produces CAVALRY. |
| WORKSHOP | Produces CATAPULT, DRAGON and ARCHER. The only building that can produce multiple unit types. |
| GUARD TOWER | A building that can attack enemies, but cannot move. |

Table 1: The list of the building units available in the game.

| Unit name | Description |
|---|---|
| PEASANT | Gathers minerals and constructs buildings, not good at fighting. |
| SPEARMAN | Effective against cavalry. |
| SWORDMAN | Effective against spearmen. |
| CAVALRY | Effective gainst swordmen. |
| DRAGON | Can fly over obstacles, can only be attacked by archers and towers. |
| ARCHER | Great counter unit against dragons. |
| CATAPULT | Easily demolishes buildings. |

Table 2: The list of the army units available in the game.

| Action Type | Action Output | Input Features |
|---|---|---|
| IDLE | NULL | NULL |
| CONTINUE | NULL | NULL |
| GATHER | resource_id | resources_features |
| ATTACK | enemy_unit_id | enemy_units_features |
| TRAIN UNIT | unit_type | unit_type_features |
| BUILD BUILDING | unit_type, (x,y) | unit_type_features, map_cells_features |
| MOVE | (x,y) | map_cells_features |

Table 3: We implement a separate action classifier per action type, because each action type needs to model a probability distribution over different objects (Action Output). For example, for the ATTACK action we need estimate a probability distribution over all visible enemy units and predict an enemy unit id, or BUILD BUILDING action needs to model two probability distributions, one over building type to be constructed, and another over all possible $(x, y)$ discrete location on the map where the future building will be placed.

and ask two humans to play the game collaboratively against a rule-based AI opponent. Both player have the same access to the game observation, but different control abilities.

The *instructor* control interface allows the human player to perform the following actions:

- **Issue** a natural language instruction to the *executor* at any time of the game. We allow any free-form language instruction.

- **Pause** the game flow at any time. Pausing allows the *instructor* to analyze the game state more thoroughly and plan strategically.

- **Warn** the *executor* player in case they do not follow issued instructions precisely. This option allows us to improve data quality, by filtering *executor*s who do not follow instructions.

On the other hand, the *executor* player gets to:

- **Control** the game units by direct manipulation using computer's input devices (e.g. mouse). The *executor* is tasked to complete the current instruction, rather than to win the game.

- **Ask** the *instructor* for either a new instruction, or a clarification.

Each human workers is assigned with either the *instructor* or the *executor* role randomly, thus the same person can experience the game on both ends over multiple attempts.

## C.1 Quality control

To make sure that we collect data of high quality we take the following steps:

**Game manual**   We provide a detailed list of instructions to a human worker at the beginning of each game and during the game's duration. This manual aims to narrate a comprehensive overview various game elements, such as player roles, army and building units, control mechanics, etc. We also record several game replays that serve as an introductory guideline to the players.

**Onboarding**   We implement an onboarding process to make sure that novice players are comfortable with the game mechanics, so that they can play with other players effectively. For this, we ask a

| Strategy Name | Description |
|---|---|
| SIMPLE | This strategy first sends all 3 initially available PEASANTs to mine to the closest resource, then it chooses one army unit type from SPEARMAN, SWORDMAN, CAVALRY, ARCHER, or DRAGON, then it constructs a corresponding building, and finally trains 3 units of the selected type and sends them to attack. The strategy then continuously maintains the army size of 3, in case an army unit dies. |
| MEDIUM | Same as SIMPLE strategy, only the size of the army is randomly selected between 3 and 7. |
| STRONG | This strategy is adaptive, and it reacts to the opponent's army. In particular, this strategy constantly scouts the map using one PEASANT and to lean the opponent's behaviour. Once it sees the opponent's army it immediately trains a counter army based on the attack graph (see Fig. 1). Then it clones the MEDIUM strategy. |
| SECOND BASE | This strategy aims to build a second TOWN HALL near the second closest resource and then it uses the double income to build a large army of a particular unit type. The other behaviours is the same as in the MEDIUM strategy. |
| TOWER RUSH | A non-standard strategy, that first scouts the map in order to find the opponent using a spare PEASANT. Once it finds it, it starts building GUARD TOWERs close to the opponent's TOWN HALL so they can attack the opponent's units. |
| PEASANT RUSH | This strategy sends first 3 PEASANTs to mine, then it keeps producing more PEASANTs and sending them to attack the opponent. The hope of this strategy is to beat the opponent by surprise. |

Table 4: The rule-based strategies we use as an opponent to the human players during data collection.

novice player to perform the *executor*'s duties and pair them with a bot that issues a pre-defined set of natural language instructions that implements a simple walkthrough strategy. We allocate enough time for the human player to work on the current instruction, and to also get comfortable with the game flow. We let the novice player play several games until we verify that they pass the required quality bar. We assess the performance of the player by running a set of pattern-matching scripts that verify if the performed control actions correspond to the issued instructions (for example, if an instruction says "build a barrack", we make sure that the player executes the corresponding low-level action). If the human player doesn't pass our qualification requirements within 5 games, we prevent them from participating in our data collection going forward and filter their games from the dataset.

**Player profile**   We track performance of each player, breaking it down by a particular role (e.g. *instructor* or *executor*). We gather various statistics about each player and build a comprehensive player profile. For example, for the *instructor* role we gather data such as overall win rate, the number of instructions issued per game, diversity of issued instructions; for the *executor* role we monitor how well they perform on the issued instruction (using a pattern matching algorithm), the number of warnings they receive from the *instructor*, and many more. We then use this profile to decide whether to upgrade a particular player to playing against stronger opponents (see Appendix C.2) in case they are performing well, or prevent them from participating in our data collection at all otherwise.

**Feedback**   We use several initial round of data collection as a source of feedback from the human players. The received feedback helps us to improve the game quality. Importantly, after we finalize the game configuration, we disregard all the previously collected data in our final dataset.

**Final filtering**   Lastly, we take another filtering pass against all the collected game replays and eliminate those replays that don't meet the following requirements:

- A game should have at least 3 natural language instructions issued by the *instructor*.
- A game should have at least 25 low-level control actions issued by the *executor*.

By implementing all the aforementioned safe guards we are able to gather a high quality dataset.

(a) Top 500 most frequent instructions      (b) Top 500 most frequent words

Figure 2: Frequency histograms for the dataset instructions and words.

## C.2 Rule-based bots

We design a set of diverse game strategies that are implemented by our rule-based bots ( Table 4). Our handcrafted strategies explore much of the possibilities that the game can offer, which in turn allows us to gather a multitude of emergent human behaviours in our dataset. Additionally, we employ a resource scaling hyperparameter, which controls the amount of resources a bot gets during mining. This hypermarameter offers a finer control over the bot's strength, which we find beneficial for onboarding novice human players. We pair a team of two human players (the *instructor* and *executor*) with a randomly sampled instance of a rule-based strategy and the resource scaling hyperparameter during our data collection, so the human player doesn't know in advance who is their opponent. This property rewards reactive players. We later observe that our models are able to learn the scouting mechanics from the data, which is a crucial skill to be successful in our game.

# D  Model architecture

## D.1 Convolutional channels of Spatial Encoder

We use the following set of convolutional channels to extract different bits of information from spatial representation of the current observation.

1. **Visibility**: 3 binary channels for each state of visibility of a cell (VISIBLE, SEEN, and INVISIBLE).

2. **Terrain**: 2 binary channels for each terrain type of a cell (grass or water).

3. **Our Units**: 13 channels for each unit type of our units. Here, a cell contains the number of our units of the same type located in it.

4. **Enemy Units**: similarly 13 channels for visible enemy units.

5. **Resources**: 1 channel for resource units.

| Linguistic Phenomena | Example |
|---|---|
| Counting | *Build 3 dragons.* |
| Spatial Reference | *Send him to the choke point behind the tower.* |
| Locations | *Build one to the left of that tower.* |
| Composed Actions | *Attack archers, then peasants.* |
| Cross-instruction anaphora | *Use it as a lure to kill them.* |

Table 5:  Complex linguistic phenomena emerge as humans instruct others how to play the game.

## D.2 Action Classifiers

At each step of the game we predict actions for each of the player's units, we do this by performing a separate forward pass for ofv the following network for each unit. Firstly, we run an MLP (Fig. 3) based action classifier to sample the unit's ACTION TYPE. We feed the unit's global summary features (see Fig. 3 of the main paper) into the classifier and sample an action type (see Table 3 for the full list of possible actions). Then, given the sampled action type we predict the ACTION OUTPUT based on the unit's features, unit dependent instructions features, and the action input features. We provide an overview of ACTION OUTPUTs and INPUT FEATURES for each actions in Table 3. In addition, you can refer to the diagram Fig. 4.

## E   Dataset details

Through our data collection we gather a dataset of over 76 thousand of instructions and corresponding executions. We observe a wide variety of different strategies and their realizations in natural language. For example, we observe emergence of complicated linguistic constructions (Table 5).

We also study the distribution of collected instructions. While we notice that some instructions are more frequent than others, we still observe a good coverage of strategies realizations, which serve as a ground for generalization. In Table 7 we provide a list of most frequently used instructions, and in Fig. 2 shows the overall frequency distribution for instructions and words in our dataset.

Figure 3: The ACTION TYPE classifier is parameterized as an MLP network to model a softmax distribution over action types based on the unit's global summary features vector.

Finally, we provide a random sample of 50 instructions from our dataset in Table 6, where showing the diversity and complexity of the collected instructions.

| Instruction |
| --- |
| *Build 1 more cavalry.* |
| *Attack peaons.* |
| *Build barrack in between south pass at new town.* |
| *Have all peasants gather minerals next to town hall.* |
| *Have all peasants mine ore.* |
| *Fight u peaas.* |
| *Stop the peasants from mining.* |
| *Build a new town hall between the two west minerals patches.* |
| *Build 2 more swords.* |
| *Use cavalry to attack enemey.* |
| *Explore and find miners.* |
| *If you see any idle peasants please have them build.* |
| *Okay that doesn't work then build them on your side of the wall then.* |
| *Create 4 more archers.* |
| *Make a new town hall in the middle of all 3.* |
| *Attack tower with catas.* |
| *Kill cavalry and peasants then their townhall.* |
| *Attack enemy peasants with cavalry as well.* |
| *Send all peasants to collect minerals.* |
| *Attack enemy peasant.* |
| *Keep creating peasants and sending them to mine.* |
| *Send one catapult to attack the northern guard tower send a dragon for protection.* |
| *Send all but 1 peasant to mine.* |
| *Mine with the three peasants.* |
| *Use that one to scout and don't stop.* |
| *Bring scout back to base to mine.* |
| *You'll need to attack them with more peasants to kill them.* |
| *Build a barracks.* |
| *Send all peasants to find a mine and mine it.* |
| *Start mining there with your 3.* |
| *Make four peasants.* |
| *Move archers west then north.* |
| *Attack with cavalry.* |
| *Make two more workers.* |
| *Make 2 more calvary and send them over with the other ones.* |
| *Return to base with scout.* |
| *Build 2 peasants at the new mine.* |
| *If attacked retreat south.* |
| *Make the rest gather minerals too.* |
| *All peasants flee the enemy.* |
| *Attack the peasants in the area.* |
| *Attack the last archer with all peasants on the map.* |

Table 6: Examples of randomly sampled instructions.

| Instruction | Frequency | Instruction | Frequency |
|---|---|---|---|
| *Attack.* | 527 | *Send idle peasants to mine.* | 68 |
| *Send all peasants to mine.* | 471 | *Attack that peasant.* | 68 |
| *Build a workshop.* | 414 | *Send all peasants to mine minerals.* | 65 |
| *Retreat.* | 323 | *Build a barracks.* | 64 |
| *Build a stable.* | 278 | *Build barrack.* | 62 |
| *Send peasants to mine.* | 267 | *Return to mine.* | 62 |
| *All peasants mine.* | 266 | *Build peasant.* | 61 |
| *Send idle peasant to mine.* | 211 | *Build catapult.* | 61 |
| *Build workshop.* | 191 | *Create a dragon.* | 61 |
| *Build a dragon.* | 168 | *Mine with peasants.* | 60 |
| *Kill peasants.* | 168 | *Build 3 peasants.* | 59 |
| *Attack enemy.* | 166 | *Defend.* | 58 |
| *Attack peasants.* | 159 | *Build cavalry.* | 58 |
| *Build a guard tower.* | 146 | *Make an archer.* | 58 |
| *Attack the enemy.* | 142 | *Attack dragon.* | 58 |
| *Stop.* | 141 | *Send all peasants to collect minerals.* | 57 |
| *Attack peasant.* | 139 | *Defend base.* | 57 |
| *Kill that peasant.* | 132 | *Build 2 more peasants.* | 56 |
| *Mine.* | 119 | *Build 2 peasants.* | 55 |
| *Build another dragon.* | 113 | *Make 2 archers.* | 55 |
| *Make another peasant.* | 113 | *Make dragon.* | 54 |
| *Build stable.* | 112 | *Build 2 dragons.* | 54 |
| *Make a dragon.* | 110 | *Attack dragons.* | 54 |
| *Build a blacksmith.* | 108 | *Make a stable.* | 53 |
| *Build a catapult.* | 108 | *Make a catapult.* | 53 |
| *Back to mining.* | 106 | *Build 6 peasants.* | 52 |
| *Build another peasant.* | 104 | *Attack archers.* | 50 |
| *Make a peasant.* | 98 | *Kill all peasants.* | 50 |
| *Build a barrack.* | 97 | *Build 2 catapults.* | 50 |
| *Build 4 peasants.* | 93 | *Idle peasant mine.* | 49 |
| *Have all peasants mine.* | 92 | *Make peasant.* | 48 |
| *Build 2 archers.* | 90 | *Attack enemy peasant.* | 48 |
| *Build dragon.* | 87 | *Attack archer.* | 48 |
| *Attack with peasants.* | 87 | *Build another archer.* | 47 |
| *Return to mining.* | 87 | *Make 4 peasants.* | 47 |
| *Build a peasant.* | 86 | *Make 3 peasants.* | 47 |
| *Idle peasant to mine.* | 85 | *Build 2 more archers.* | 46 |
| *Make a workshop.* | 83 | *Send idle peasant back to mine.* | 46 |
| *Create a workshop.* | 81 | *Make more peasants.* | 46 |
| *Mine with all peasants.* | 80 | *Make 2 more peasants.* | 46 |
| *Build 3 more peasants.* | 79 | *Build blacksmith.* | 46 |
| *Create another peasant.* | 79 | *Collect minerals.* | 45 |
| *Send all idle peasants to mine.* | 77 | *Kill.* | 45 |
| *Build 3 archers.* | 77 | *Build an archer.* | 45 |
| *Kill peasant.* | 77 | *Keep mining.* | 45 |
| *Make another dragon.* | 76 | *Keep attacking.* | 43 |
| *Kill him.* | 72 | *Attack dragons with archers.* | 43 |
| *Build guard tower.* | 70 | *Create a stable.* | 42 |
| *Attack town hall.* | 70 | *Make 3 more peasants.* | 42 |
| *Start mining.* | 69 | *Attack the peasant.* | 41 |

Table 7: The top 100 instructions sorted by their usage frequency.

Figure 4: Separate classifiers for each of the available action types.