[Reviews · NeurIPS 2019]

Reviewer 1



Post Rebuttal: Thank you for your response. I do see the advantages you listed to support the choice of language over programs. Overall, I feel the general direction of using language for intermediate supervision is really interesting and worthy of further study. This paper could be significantly improved however in some regards. For example: - Authors should study the generated language to test it for compositionality (as other reviewers have pointed out). One way to do this is to create an analogy task (A:B::C:D). You can create 4 settings: i) where X has to be done followed by Y, ii) Y has to be done followed by X, (ii) X has to be done followed by Z, iv) Z has to be done by X. Here X, Y, Z are some game requirements like build a hut or collect a treasure. Then if the agent has seen i-iii in the training data but not iv, it should still be able to generate the appropriate response. (You can create synthetic data for this task to do a controlled study). - Human evaluation would also be useful. - Please list many examples of generated response in the paper. You can also list the k-nearest neighbor in the training data to show that significantly new response are being generated. You can use BERT or some other embedding for similarity. ----------------- This paper proposes an interesting direction of using latent language for guiding agent's in multi-agent strategy game. They propose a model that consists of an instruction generator and an executor. Given an encoding f(s) of the current state s, the model first decides to generate an instruction or not. The set of generated instructions along with f(s) is passed to the executor to take actions. This is similar to Hierarchical Reinforcement Learning (HRL) except instead of macro-actions we have text. The idea is that text is more expressive than macro-actions. The system is trained using supervised learning by collecting labeled data of instructions and trajectory using a set of human players competing against rule-based agents. The game is a multi-agent strategy game where resources have to be optimized, enemy armies need to be defeated etc. The game consists of simple images and 2 players. I believe the evaluation is done against rule-based bots. Strengths: - Systems with latent language are an interesting direction - Latent language is more expressive than macro-actions - Latent language provides interpretability Weakness: - Latent language seems to add a more complex intermediate problem. You are now introducing text understanding which might be a harder problem. Do we really need text? why not use a program for guidance? Surely, a program is more expressive than macro-action and interpretable and you dont have language understanding challenges. Maybe a clever data collection strategy can collect programs. - The problem is solved using supervised learning without any exploration based learning. This makes me wonder how easy the setup is. Did you try comparing against agents that are trained without text but use reinforcement learning? The environment must have some reward (score, number of enemies killed etc.). Of course, once you consider exploration it is not clear how accurate your instruction model would be. Maybe this is a limitation of this direction? Questions and Other Comments - Why would any instruction be repeated at all? I am trying to understand the purpose of using one-hot vector encoding for instructions. How many instructions occur more than once? - Did you evaluate against rule-based bots? How is the performance against another model trained using the same strategy (i.e. self play). - I believe authors first generate a text using the instruction model and then re-encode the text using an encoder. Why not do something like this: create an instruction embedding g(f(s)) from state encoding f(s). Pass this instruction encoding directly to the executor (as opposed to generated text). Then you can add an auxiliary objective which will try to bring g(f(s)) closer to the gold instruction encoding. This might work better as you are not adding discretization in between which can fail due to a single wrong decoding (say as based on a tie). - Equation on line 214 should have different state s for each time "i". Otherwise your state representation is not changing while you take new actions. - Many missing citations (see Tellex et al., AAAI 2011, Tellex et al., RSS 2014, Chaplot et al., AAAI 2017, Bahdanau et al., ICLR 2017, Misra et al., EMNLP 2018, Mirowski et al., 2019, Chen et al., CVPR 2019) etc.

Reviewer 2



The term "latent natural language instructions" appears many times throughout the paper. The problem is that the instructions aren't latent: they're provided as supervised training data, and modeled as such. See the Andreas paper Learning with Latent Language (which you might want to cite) for why the language signal is latent in their approach. A similar comment applies to the idea of a "latent plan". Since the plans are represented in natural language, and provided as supervised training data, the plans aren't latent either. I liked the fact that the instructions "contain a number of challenging linguistic phenomena", and the examples provided clearly showcase this aspect of the dataset. Overall the paper is well-written, but there are some non-native phrasings which could be improved for the next version, eg "If an enemy unit goes out the players visibility"; "embed them in isolation with the 4.1.3." (can't use a section number like this) It would be good to have more details of the RNN encoder. Is it a vanilla RNN, LSTM? Overall there were a number of aspects of the experimental setup and the agent architecture where I had to work hard to ascertain the details. One example is the fixed set of instructions, and where this set comes from. The caption in Table 3 says N most frequent. This is a detail that needs stating earlier in the paper and in the body of the text. As far as I can tell, the RNN is never used to autoregressively generate instructions, eg using a beam search at decoding time: the generative model is only ever used to rank a subset of the instructions from the training data. It would be interesting to see what happens when the agent can generate instructions, especially some not seen in the training data. It would also be good to see some examples of such generated/sampled instructions (perhaps in the appendix). It would be useful to at least have some idea in the main body of the paper what the rule-based bots are like and how they are implemented. Section 5.3 has very little content. This space could be used more productively, either to provide some of the details alluded to above, or to provide a more informative analysis. The suggestion for future work of using RL and generating novel instructions is intriguing. Either would improve the current paper, although the use of RL could reasonably be left for another paper I think. The generation of novel instructions might go in this version, and if not I would be clear early on in the paper that the generative models are only being used to rank the set of instructions from the training set. Overall this is a promising paper, with an interesting and potentially useful dataset, but which isn't quite ready for a top-tier ML conference such as NeurIPS. Comment after author response: thanks for reading the reviews and responding accordingly. The authors comment that the language is clearly latent at test time, but this has to be true, otherwise there's nothing to predict (as far as the plan is concerned). My understanding of "latent" is that it typically refers to a random variable which is not observed during training.

Reviewer 3



This work introduces a dataset built on top of MiniRTS which aims to use “natural language” as the communication paradigm for providing and following instructions. Agents can also choose to ignore the instruction and act based on the environment. A large scale dataset based on human instructions is collected to fascilitate training and collected based on play against a rule based agent. There are a series of architectural choices that are than made for each of the actions and a series of encoder/decoder network choices which are compared for the communication protocol.

[Author Response · NeurIPS 2019]

We would like to thank all the reviewers for their insightful and constructive feedback. We are glad that they liked our framework of using natural language for planning, our environment, and our large-scale dataset.

We first address several common points:

- Reviewers were curious if we can sample novel instructions autoregressively. The RNN-Generative produces well-formed language and we can indeed use the generated instructions, instead of the pre-selected top 500 instructions, to instruct the executor. This model can get comparable win rate to the RNN-Discriminative in Table3. We will include this number and samples of generated instructions in the camera ready.

- We also want to re-emphasize evidence for the importance of the compositionality of natural language. We show this by comparing RNN/BoW models (compositional) against OneHot (non-compositional). Further, we showed that it is important to consider a sequence of history instructions in such complex context around line250. This result shows the need to compose information from across multiple instructions for good performance.

Finally, we appreciate the reviewers for suggesting additional citations and interesting future directions. We will add those in the camera ready.

**Response to Reviewer 1**

Natural language has several advantages over latent programs. Firstly, natural language is highly expressive and can be applied to many domains where actions would be difficult to represent with programs. At the least, the space of programs would likely have to be engineered for each new domain, which is not the case with natural language. Secondly, gathering supervision for natural language actions is possible with the framework we introduce.

We certainly do not claim to be "solving this task" in the paper. In Table3, the comparison is made between a hierarchical agent that uses language and an agent that does not use language. Both agents are trained on the same dataset. One of our major claims is that having such hierarchy with natural language as intermediate instructions is helpful. Training an RL agent for such RTS environment is feasible, as demonstrated by the DeepMind's effort in Starcraft II, but remains challenging and highly computationally expensive.

Many simple instructions such as "attack", and "build peasants" are very frequent, and can be used in many situations. Please see Table7 in appendix for most frequent instructions with their frequency.

We have indeed evaluated the agents against rule-based bots and the differences between different models and overall trend is similar to the results in Table3. Training with selfplay with unit-level control is challenging and beyond the scope of this paper.

We generate actions for all units at once, ignoring their orders and dependency.

**Response to Reviewer 2**

Thanks for the terminology suggestion, and the missing reference. At test time, the language is clearly latent, because it is intrinsic to the model's decision making process and has no other effects. However, at training time we rely on the supervised data to learn to use natural language. We agree that the distinction could be clearer, and will update the paper.

We have included description of the rule based bots used for collecting data in the appendix due to page limit. Please note that we do not compare our trained models against rule-based bots but rather compare models that uses language against a baseline that does not. Therefore the details on those bots are less important. The RNN used in the paper is a one layer LSTM.

**Response to Reviewer 3**

The claim for compositionality is mainly demonstrated in OneHot model (non-compositional) vs BoW/RNN models (compositional). As we can see from the both Table2 and Table3 that the compositional models dramatically outperform the non-compositional model in terms of both likelihood and win rate. In addition, although the RNN executor and BoW executor has little difference in terms of likelihood, the RNN instructor outperforms BoW instructor with a relative large margin in terms of both likelihood and win rate.

We can play the game by typing the instruction to instruct the executor. The executor responds accurately to those instructions. We can also generate instructions with trained instructor and control units ourselves through game interface.

The baseline model is trained with supervised learning while other more complex RTS agents are trained with reinforcement learning with significantly more computation resources and samples.

We believe that our method that factorizes unit actions to type and argument classifiers is more generalizable and scalable. A similar approach was also adopted by OpenAI's Dota bot trained in large scale RL setting.

[Meta-Review · NeurIPS 2019]

After author feedback and reviewer discussion, this paper received diverging final ratings of 7 (R1), 3 (R2) and 4 (R3). Given this lack of consensus, the AC read the paper, reviews, feedback, and discussion closely, and in this case decided to accept. Of the three reviews, R2’s rating was the most negative. R2’s review noted that the paper was well-written, and praised the inclusion of challenging linguistic phenomena in the dataset, while raising concerns with the characterization of the language as ‘latent’ and requesting additional details (e.g. regarding the RNN encoder and the ranking approach). In the context of the review itself, R2’s rating (3, ‘clear reject’) appears to be calibrated to a relatively strict standard, possibly stricter than some other NeurIPS reviewers. Turning to the paper itself, in the opinion of the AC, the use of natural language to decompose complex tasks into manageable subgoals is an important research direction that is worthy of further study. As noted in author feedback and by R1, natural language has some advantages over program specifications, e.g. interpretability, applicability to multiple tasks, free availability in some cases. In this paper, the authors show in (a real-time strategy game) that models that are trained to generate and then follow natural language instructions outperform models trained only to directly imitate human actions. This is not a trivial finding, and has not been widely demonstrated. The paper is clearly written, and comes with code and a detailed supplementary. Therefore, it is the opinion of the AC that this paper would be of significant interest to the NeurIPS community, and groups working on hierachical RL, options frameworks, hindsight experience replay, etc. Having said that, R2 and R3 raise valid concerns. R2 correctly notes that a variable which is explicitly annotated is not latent. R3 raises concerns regarding the claim that the increase in performance can be attributed to the compositionality of language. It is the opinion of the AC that these issues can be addressed in the camera-ready version (and should be). Regarding compositionality, it might be advisable to make a weaker claim that does not specifically invoke compositionality (which is not studied in detail in the paper).